# Influence of Fetomaternal Microchimerism on Maternal NK Cell Reactivity against the Child’s Leukemic Blasts

**DOI:** 10.3390/biomedicines10030603

**Published:** 2022-03-04

**Authors:** Lena-Marie Martin, Anne Kruchen, Boris Fehse, Ingo Müller

**Affiliations:** 1Research Institute Children’s Cancer Center Hamburg, 20246 Hamburg, Germany; l.m.martin@gmx.net; 2Division of Pediatric Stem Cell Transplantation and Immunology, Clinic of Pediatric Hematology and Oncology, University Medical Center Hamburg-Eppendorf, Martinistr. 52, 20246 Hamburg, Germany; a.kruchen@uke.de; 3Research Department Cell and Gene Therapy, Department of Stem Cell Transplantation, University Medical Center Hamburg-Eppendorf, 20246 Hamburg, Germany; fehse@uke.de

**Keywords:** fetal microchimerism, haploidentical stem cell transplantation, KIR, HLA, NK cell alloreactivity

## Abstract

Persistence of fetal cells in the circulation of the mother (fetal microchimerism, FM) is associated with increased survival and reduced relapse of children with leukemia receiving a haploidentical hematopoietic stem cell transplantation (hHSCT). NK cells play an important role in maternal tolerance towards the unborn child. In this study, 70 mother–child pairs were prospectively analyzed for the occurrence of FM, KIR genotype and HLA-C type. We found that occurrence and level of FM were influenced by three maternal genetic factors: presence of an HLA-C1 allele, absence of KIR2DL3 and presence of a cen-B/B motif. Furthermore, an HLA-C match between mother and child favored persistence of FM. NK cells from FM^+^ mothers showed a 40% higher specific degranulation against their filial leukemic blasts than NK cells from FM^−^ mothers, suggesting the presence of educated maternal NK cells. Nevertheless, cytotoxicity of parental NK cells against filial leukemic blasts was independent of KIR genetics (haplotype, B content score, centromeric and telomeric KIR gene regions) and independent of FM, indicating that additional immune effector mechanisms contribute to the beneficial effect of persisting FM in hHSCT.

## 1. Introduction

Haploidentical hematopoietic stem cell transplantation (hHSCT) is well established for pediatric patients lacking an HLA-identical sibling and a matched-unrelated donor [1,2]. Depending on the donor, differences in outcome have been described: in non-T-cell-depleted hHSCT, the father is the better donor, whereas in T-cell-depleted hHSCT the mother is the better donor [3]. T-cell-depleted mother-to-child transplantations were associated with less relapse and increased event-free survival compared to father-to-child transplantations [4,5]. Fetomaternal microchimerism (FM) is established in virtually all mothers during pregnancy, when cross-placental cell traffic occurs between maternal and filial cells [6]. We showed that persistent fetal microchimerism (FM^+^) in mothers, who donated stem cells for their children, was associated with an increased overall survival and reduced incidence of relapse, compared to father-to-child transplantations and transplantations from mothers without FM (FM^−^) [7]. During establishment of FM, a special subtype of NK cells, the decidual NK (dNK) cells, play a crucial role [8]. dNK cells show an altered receptor repertoire marked by increased expression of killer immunoglobulin-like receptors (KIRs). These interact with their respective receptors on the trophoblast and were shown to be crucial for placentation and successful pregnancy progression [9,10,11,12,13,14,15]. 

NK cells are important not only during pregnancy, but also after hematopoietic stem cell transplantation (HSCT). NK cells have the ability to recognize degenerated and virus infected cells by loss or down-regulation of HLA molecules [16,17,18] and are the first lymphocytes to reconstitute after HSCT [19]. Hence, they are the first effector cells that might exert a beneficial graft-versus-leukemia (GvL) effect [20,21]. NK cells strongly interact with HLA molecules by KIR receptors. The human KIR family consists of 15 KIR genes. KIR receptors are divided into either activating or inhibitory receptors, with the exception of KIR2DL4, whose mode of action is depending on the ligand [22]. Most KIR haplotypes can be categorized either into haplotype A or haplotype B [23]. Haplotype A is characterized by allelic variation, while haplotype B is marked by varying KIR gene content [24,25]. Furthermore, KIR genes can be grouped depending on their centromeric and telomeric KIR gene motifs [26]. Based on the number of activating KIR genes the B content score can be determined. KIR genes, which are located on chromosome 19, are highly polymorphic and polygenic and are inherited independently of their HLA ligands located on chromosome 6 [27]. Consequently, each individual carries a unique KIR/HLA combination [28].

During an educational process, NK cells learn to distinguish between self and non-self, depending on KIR receptor engagement [29,30,31]. After HSCT, educated NK cells can be alloreactive against cells lacking the respective HLA ligand. Hence, potential NK cell alloreactivity can be predicted for mismatched HSCT. The main KIR/HLA interactions considered here are: KIR2DL1/HLA-C2, KIR2DL2/L3/HLA-C1 and KIR3DL1/HLA-Bw4. 

Activating KIR genes in donors with a cen-B/B motif are associated with a better outcome in a cohort of adult and pediatric AML patients undergoing unrelated HSCT [32]. Moreover, the presence of KIR2DS1 in donors is associated with a better outcome and protection from relapse [33]. In addition, the presence of centromeric KIR B genes in the donor is associated with increased survival in HLA-identical sibling transplantations [34] as well as with a reduced relapse risk and prolonged event-free survival (EFS) in pediatric ALL patients receiving a matched unrelated HSCT [35]. Moreover, a significantly reduced risk of relapse and better EFS was ascribed to KIR B haplotype donors in pediatric ALL patients receiving a hHSCT [36].

Here, we investigated the role of NK cells in maintaining FM on the one hand and the effect of FM on the reactivity of maternal NK cells against leukemic blasts of their children on the other hand. To this end, we analyzed the influence of maternal and filial HLA/KIR genes on the occurrence and level of FM and the impact of persisting FM on maternal NK cell KIR phenotype. Moreover, the functionality of maternal NK cells against their child’s leukemic blasts was determined in vitro.

## 2. Materials and Methods

### 2.1. Study Population

Seventy parent–child pairs were enrolled. At the time of blood donation, all children were being treated for a hematological disease at the Clinic of Pediatric Hematology and Oncology, University Medical Center Hamburg-Eppendorf. All participants gave their written consent in accordance with the Declaration of Helsinki. Ethical approval was given by the local ethics committee (approval No. PV4296).

### 2.2. Cell Isolation and Treatment

PBMCs were isolated from peripheral blood and leukemic blasts from bone marrow (if not otherwise stated) using Biocoll separation solution. NK cells were isolated from PBMCs using the NK cell isolation kit (Miltenyi Biotec, Bergisch Gladbach, Germany) following the manufacturer’s recommendations and pre-activated overnight at 1 × 10^6^ cells/mL in RPMI1640 supplemented with 10% autologous plasma, 1% penicillin/streptomycin (ThermoFisher Scientific, Schwerte, Germany), 100 U/mL IL-2 (ImmunoTools, Friesoythe, Germany) and 20 ng/µL IL-15 (Miltenyi).

### 2.3. DNA Isolation and Genotyping

Genomic DNA was prepared either directly from 450 µL peripheral blood or from 5–10 × 10^6^ PBMCs or leukemic blasts using the QIAamp DNA Blood and Tissue Kit (QIAGEN, Hilden, Germany). As described previously, HLA-C genotyping was performed by PCR [37], and KIR genotyping by real-time PCR [38]. Individuals were grouped depending on their KIR gene motifs as described by Cooley et al. [32].

### 2.4. Microchimerism Determination

For microchimerism analysis, markers were selected for each mother/child pair using a panel of short insertion/deletion polymorphisms (InDels) [39], as well as a Y-chromosome-specific sequence [40]. Marker selection was performed as described previously [7]. A TaqMan-based digital droplet PCR (ddPCR) was performed to quantify the level of fetal cells with the QX200 Droplet Digital PCR System (Bio-Rad Laboratories, Hercules, CA, USA) following the manufacturer’s recommendations. Briefly, two reactions were set up: an analysis reaction and a quantification reaction. A final ddPCR reaction mixture of 22 µL total volume consisted of 11 µL 2× ddPCR Supermix (Bio-Rad Laboratories), 900 nmol/L of the two primers and 250 nmol/L of the probe (FAM-BHQ1 labeled) for the sequence of interest and 0.5 µL FastDigest either *EcoR*I (ThermoFisher, Scientific, Waltham, MA, USA) or if a restriction site for *EcoR*I was detected in the amplicon, *HinD*III was used (ThermoFisher, Scientific, Waltham, MA, USA). Marker-specific amounts of MgCl_2_ were added. In the analysis reaction, at least 2 µg genomic DNA was distributed in 500 ng DNA/well. To verify DNA concentration, the quantification reaction consisted of 100 ng DNA/well. Additionally, 900 nmol/L of two primers and 250 nmol/L of a probe (HEX-BHQ1 labeled) detecting the human hematopoietic cell kinase gene were added, for determination of the total amount of input DNA. The PCR mixture was compartmentalized using the QX200 Droplet Generator (Bio-Rad). The generated water–oil emulsion droplets were transferred to a 96-well PCR plate (Eppendorf, Hamburg, Germany), and amplified in a thermal cycler with the following conditions: denaturation (95 °C for 10 min), amplification (94 °C for 30 s, 60 °C for 1 min, 40 cycles), ramp rate of 1.5 °C/s, followed by a final 98 °C activation step for 10 min. Afterwards, droplets were analyzed for FAM and HEX signals simultaneously using the QX200 droplet reader following the manufacturer’s instructions. Data were processed using the QuantaSoft software (Bio-Rad Laboratories) and the mean concentration of the target sequence (copies per 10 µL) was calculated by Poisson Statistics. To determine unspecific background, the same number of reactions was prepared with DNA from a control cell line negative for the respective marker. Level of FM was calculated as copies per 100,000 background copies considering haploidy (Y-chromosomal) or diploidy of the respective marker. Samples were considered as positive (FM^+^) if the detected FM was higher than the measured background level in the negative cell line. Mothers without detectable microchimeric cells are further depicted as FM^−^.

### 2.5. Degranulation, KIR Phenotyping and Cytotoxicity Assay

IL-15/IL-2 pre-activated parental NK cells were either cultivated with the HLA negative cell line K562 or with freshly thawed leukemic blasts from the parent’s child. Target cells were labeled with 2 mM eFluor670 (eBiosciences, San Diego, CA, USA) following the manufacturer’s instructions. After staining, cells were washed twice with assay medium (RPMI 1640 w/o phenol red supplemented with 10% FBS, 1% L-glutamine and 1% penicillin/streptomycin (all ThermoFisher, Scientific)). For cytotoxicity determination, cells were co-cultured in effector to target ratios (E:T ratios) of 20:1, 10:1 and 5:1 for four hours. Maximal lysis of cells was determined by treatment with 1% Triton X-100 30 min before analysis. Dead cells were determined by addition of propidium iodide 1:100 directly before measurement. Cytotoxicity was analyzed on a Canto I (BD Biosciences, San Jose, CA, USA) or a MACSQuant10 (Miltenyi Biotec). For analysis of degranulation, NK cells were co-cultured in an E:T ratio of 1:1. One hour after incubation, GolgiStop (BD Biosciences, San Jose, CA, USA) and CD107a-PECy7 (Biolegend, San Diego, CA, USA) were added followed by incubation for additional 3 h. After incubation, cells were washed once with FACS buffer (PBS supplemented with 2% FBS) and stained with anti-KIR antibodies provided in the supplementary. Cells were washed once with FACS buffer and fixed using 1% PFA in FACS buffer. Cells were analyzed either on a FACSCanto I (BD Biosciences, San Jose, CA, USA) or a LSRFortessa (BD Biosciences). In analogy to the well-established “specific lysis”, we defined “specific degranulation” of NK cells to make experiments comparable. Degranulation of NK cells against K562 was set as maximum degranulation, and specific degranulation was calculated by the following formula: percentage of specific degranulation = (percentage of degranulation against filial blasts x 100)/maximum degranulation.

### 2.6. Statistical and Flow Cytometry Data Analysis

Statistical analysis of correlations with Fisher’s exact test for two groups, or Chi^2^ for more than two groups, was performed using IBM SPSS Statistics Version 26 (IBM Corp.). The Mann–Whitney test was used for comparison of two groups; the Kruskal–Wallis test followed by Dunn’s multiple comparison test was used for more than two groups using GraphPad PRISM Version 8.4.1. Analysis of FACS data was conducted with FlowJo V10.6.3 or BD FACSDiva Version 8.0. Significance levels are highlighted by * for *p* < 0.05, ** for *p* < 0.01, *** for *p* < 0.005 and **** for *p* < 0.001.

## 3. Results

### 3.1. Study Cohort

Seventy mothers whose children were treated for a hematological disease at the University Medical Center Hamburg-Eppendorf were analyzed for FM in peripheral blood. Patient characteristics are summarized in Table 1. Proportions of male and female patients were equal. The median age was 5.72 years and 58 of the children suffered from acute leukemias: c-ALL, other B-precursor ALLs, T-ALL and AML. Twelve children suffered from various other diseases, namely T-cell non-Hodgkin lymphoma (T-NHL) (*n* = 2), hemophagocytic lymphohistiocytosis (HLH) (*n* = 1), neuroblastoma (NB) (*n* = 4), mucopolysaccharidosis type I (MPSI) (*n* = 1), and immune dysregulation polyendocrinopathy enteropathy X-linked (IPEX) (*n* = 1). Maternal DNA was analyzed for the occurrence of FM using digital droplet PCR (ddPCR). Differentiation of maternal and filial DNA was performed by detection of short insertion/deletion (InDels) polymorphisms [40,41]. A suitable InDel was found in 73% of the mother–child pairs and FM was detected in 37% of these mothers (Table 1).

### 3.2. Effect of Filial Age and Sex on FM 

We sought to identify factors affecting the occurrence and the level of FM. Detailed statistical analyses are provided in Appendix A. The youngest child in the cohort was 2 weeks and the oldest child was 17.6 years of age at the time of analysis. FM was detected in 52% of mother–daughter pairs compared to 25% of mother–son pairs (HR 3.989, *n* = 51, *p* = 0.080; Cramer-V 0.280; Figure 1A). The level of FM was significantly higher in mother–daughter pairs compared to mother–son pairs (mean FM cells: daughter 7.53 vs. son 1.45 per 100,000 PBMC; *p* = 0.029; Figure 1B). FM was detected in 36% of the mothers with children between 0 and 6 years, in 33% of the mothers with children between 7 and 12 years and in 50% of the mothers with a child older than 12 years (Figure 1C). With increasing children’s age, the amount of microchimeric cells slightly decreased, but linear regression analysis showed that children’s age and FM were not correlated (r^2^ = 0.003) (Figure 1D). Taken together, the influence of filial age and sex on the occurrence of FM did not reach statistical significance, but the level of FM was significantly affected by filial sex.

### 3.3. Influence of HLA-C on FM

We hypothesized that HLA-C and KIR constellations might have an impact on the persistence of FM. We grouped child, mother, and father depending on their HLA-C genotype into C1 and C2 homozygous and C1/C2 heterozygous according to Paximadis et al. [42]. A significant correlation was observed between persistence of FM and maternal HLA-C genotype (6.979, *n* = 51, *p* = 0.031, Cramer-V 0.387) (Table 2).

A more in-depth view on the frequencies revealed that FM was detected only in mothers carrying at least one HLA-C1 allele (FM^+^: C1/C1 53% vs. C1/C2 47% vs. C2/C2 0%) (Figure 2A). No FM was detected in C2 homozygous mothers. Additionally, we found that a significantly higher level of FM was detected in C1 homozygous mothers with a C1 homozygous child compared to C1/C2 heterozygous mothers with a C1 homozygous child (mean FM cells: 12.2 vs. 0.6; *p* = 0.039) (Figure 2B). Therefore, maternal and filial HLA-C match was analyzed in more detail. FM significantly correlated with mother-to-child HLA-C match (9.188; *n* = 51, *p* = 0.009, Cramer-V 0.434), as FM was detected two times more often in HLA-C matched mother–child pairs (match: FM^+^ 79% vs. FM^−^ 34%) compared to mismatched mother–child pairs (mismatch: FM^+^ 21% vs. FM^−^ 66%) (Figure 2C). This was paralleled by a significantly higher level of FM in HLA-C matched mother–child pairs compared to mismatched pairs (mean FM cells: 7.40 vs. 0.98; *p* = 0.002) (Figure 2D). Subsequently, mismatched mothers were grouped into non-self or missing-self from the mother-to-child direction after the criteria depicted in Appendix A. In a non-self mismatch, the mother is either C1 or C2 homozygous and the child C1/C2 heterozygous. In both cases, filial cells express a non-self molecule. In the missing-self direction, the mother is heterozygous and the child homozygous. Hence, a ligand, to which the parental NK cells are educated, is absent (missing) in the child. In both mismatched settings, FM was detected at similar frequencies (FM^+^: non-self 11% vs. missing-self 19%) (Figure 2E). The level of FM was significantly increased in the HLA-C1 matched pairs compared to either non-self (mean FM cells: 12.17 vs. 0.12; *p* = 0.0075) or missing-self mismatched pairs (mean FM cells: 12.17 vs. 1.57; *p* = 0.0305) (Figure 2F).

### 3.4. Influence of KIR Genotype on FM

Since the presence of a maternal HLA-C1 appeared to favor the persistence of FM, we investigated the influence of the respective KIR receptors on the establishment and persistence of FM. Of all KIR genes analyzed, only KIR2DL3 showed a significant correlation with the occurrence of FM (*p* = 0.04, odds ratio 6.923, 95%CI 1.23–35.956) (Appendix A). Comparison of FM frequencies in KIR2DL3^+^ and KIR2DL3^−^ mothers revealed that FM was detected in 75% of KIR2DL3^−^ mothers compared to only 30% of KIR2DL3^+^ mothers (Figure 3A). Subsequently, we analyzed the influence of all KIR genes on the level of FM. Again, KIR2DL3 was the only KIR gene that correlated with the level of FM (Figure 3B; *p* = 0.009). Filial KIR genes had no effect on the level of FM in the mothers (data not shown). KIR2DL2, KIR2DL3 and KIR2DS2 all share the same ligands with high affinity for HLA-C1 and with low affinity for HLA-C2. KIR2DL3–C1 interaction results in a lower inhibition compared to KIR2DL2–C1 interaction. Absence of KIR2DL3 resulted in a significantly higher level of filial cells compared to the groups in which KIR2DL3 was present (mean FM cells 0.52 vs. 14.97; *p* = 0.0183) (Figure 3C).

Individuals can be grouped depending on their centromeric or telomeric KIR gene motifs depicted in Figure 3G. A significant correlation of FM and the maternal centromeric KIR genes (8.551, *n* = 51, *p* = 0.011; Cramer-V = 0.422) with a medium strength was detected. In the FM^−^ group, no difference in the frequencies of centromeric KIR gene motifs was seen in mother and child (Figure 3D). In the FM^+^ group, the frequency of mothers with a B/B motif was increased (B/B: FM^−^ 6% vs. FM^+^ 37%), while frequencies of A/A and A/B were decreased (A/A: FM^−^ 34% vs. FM^+^ 6% and A/B: FM^−^ 60% vs. FM^+^ 53%). Analysis of the telomeric KIR gene motifs in mothers showed that mothers with A/A and A/B genotype were almost equally distributed between the FM^+^ and FM^−^ groups (A/A: FM^−^ 53% vs. FM^+^ 42% and A/B: FM^−^ 44% vs. FM^+^ 53%), as shown in Figure 3E. Furthermore, we investigated the influence of maternal KIR gene motifs on the level of FM. A significantly higher level of FM was detected in mothers with a centromeric B/B motif compared to mothers with a centromeric A/A motif (mean FM cells: B/B 13.67 vs. A/A 0.38; *p* = 0.0314) (Figure 3F).

### 3.5. Influence of FM on Parental NK Cell KIR Phenotype

To analyze if a persistent FM in the maternal circulation leaves an imprint in the maternal KIR repertoire or if a distinct KIR repertoire supports persistence of FM, the KIR phenotype and degranulation of pre-activated maternal and paternal NK cells were assessed. KIR phenotyping was performed to detect potential changes in KIR receptor expression after co-culture with leukemic blasts or with K562. Frequencies of activated cells were determined based on expression of the degranulation marker CD107a. Frequencies of all KIR-expressing cells (Figure 4A), and activated CD107a^+^ KIR expressing cells (Figure 4B) were compared between FM^+^ and FM^−^ mothers against their children’s leukemic blasts. The general KIR expression pattern was similar in both groups (Figure 4A), but in the FM^+^ group, significantly more activated NK cells expressed KIR2DL1/S1 (*p* = 0.023), KIR2DL1/S5 (*p* = 0.051), KIR2DL3 (*p* = 0.032), and KIR2DL5 (*p* = 0.047) as compared to the FM^−^ group.

### 3.6. Influence of FM on Parental NK Cell Reactivity

NK cell reactivity against K562 and the respective filial leukemic blasts was compared between FM^+^ and FM^−^ mothers (Figure 5A). In analogy to the well-established “specific lysis”, we defined “specific degranulation” of NK cells to make experiments comparable. Degranulation of NK cells against K562 was set as maximum degranulation. NK cells from FM^+^ mothers showed a significantly higher specific degranulation compared to NK cells from FM^−^ mothers (FM^+^ 88% vs. FM^−^ 48%; *p* = 0.0098). Yet, this higher reactivity of NK cells obtained from FM^+^ mothers did not translate into higher specific lytic activity of the NK cells against leukemic blasts (FM^+^ 30.89% vs. FM^−^ 27.27% vs. father 22.12%) as shown in Figure 5B.

NK cell cytotoxicity was further analyzed according to KIR/HLA interactions, as a lack of HLA or mismatch results in alloreactive NK cells. We found that specific lysis of leukemic blasts was independent of HLA-C match or mismatch (Figure 6A,B). Although mean specific lysis of mothers with a centromeric B/B motif was increased by 15% (B/B 39% vs. A/B 24% and A/A 25%), this was not statistically significant (Figure 6D). Specific lysis was also not influenced by the telomeric motifs (A/A 29% vs. A/B 23%). While KIR2DL3 absence favored the FM, absence of KIR2DL3 or of KIR2DL2 had no significant effect on NK cell reactivity against filial leukemic blasts (KIR2DL3^−^ 43% vs. KIR2DL3^+^ 30%). However, when both receptors were present, a decreased specific lysis of leukemic blasts to 24% was detected in mothers (Figure 6C).

## 4. Discussion

In this study, mother–child pairs were analyzed for the occurrence of FM, and we assessed if certain maternal and/or filial genetic factors affect the level of FM. Furthermore, we investigated whether FM leaves an imprint on the maternal KIR phenotype and whether FM as well as different genetic factors affect parental NK cell reactivity against filial leukemic blasts.

### 4.1. NK Cells in Establishment and Persistence of FM

It still remains to be elucidated why FM persists only in some but not all mothers. In this study, FM was detected by ddPCR in 37% of the analyzed mothers. ddPCR allowed for exact quantification of fetus-derived cells. FM was observed to be independent of children’s sex and age (Table 1 and Figure 1), suggesting that fetal cells, if not eradicated soon after parturition, stably persist in the maternal circulation. Tolerogenic mechanisms allowing fetal cells to survive in the mothers are poorly understood. We found that HLA-C identity of mother and child significantly correlated with the perpetuation of FM (Table 2). Besides the favorable HLA-C match, the presence of at least one HLA-C1 allele in the mother resulted in an increased level of FM (Figure 2), even though most of the filial cells are cleared by the maternal immune system after parturition [43]. During pregnancy, the maternal immune system is suppressed to not attack the partially allogeneic fetus, which is maintained by various mechanisms, e.g., production of immunosuppressive molecules, activation of immunosuppressive cells, exclusion of immune cells and a silencing of potentially reactive T cells [44,45,46]. We speculated that distinct HLA/KIR interactions, which proved to be important for a successful pregnancy progression [14,15,47,48], are important for the tolerance of FM cells as well. However, neither presence nor absence of any of the maternal activating KIR receptors affected maintenance and/or levels of FM (Figure 3 and Figure 4).

During pregnancy, a protective effect was observed by KIR2DS1 [10] and KIR2DS5 [49,50]. Interestingly, when the fraction of activated KIR-expressing cells was analyzed after co-culture with leukemic blasts, a significantly higher fraction of KIR2DS1 and KIR2DS5 expressing cells was activated in NK cells from FM^+^ mothers as compared to cells from FM^−^ mothers (Figure 5B). The fraction of KIR2DL1 only expressing cells was not affected, suggesting a role for KIR2DS1 and KIR2DS5 in the reactivity against the filial leukemic blasts. In pregnancy, distinct maternal KIR A haplotype/fetal HLA-C combinations cause a higher risk for complications [14,15,47,48]. All FM^+^ mothers were within the haplotype B group, suggesting that haplotype B could favor the persistence of FM, and that the presence of activating KIR genes is important. In our limited cohort, this observation did not reach statistical significance. This suggests that instead of KIR genetics, the KIR expression and potential interactions are more important for FM. Besides NK cells, Tregs are important in women during pregnancy and have been described to form a memory that favors subsequent pregnancies [51,52]. Tregs from multiparous women showed an altered gene expression profile compared to nulliparous women [53]. A protective effect of these Treg cells could also explain the tolerance of filial cells in the maternal circulation independent of maternal KIR receptors and, therefore, will need to be addressed in future research.

### 4.2. NK Cell Alloreactivity, KIR and HSCT

NK cell alloreactivity includes a Graft-versus-Leukemia effect after mismatched HSCT [20,21,54]. In T-cell-depleted HSCT, NK cell alloreactivity had a positive impact on GvHD/relapse free survival [55]. In pediatric ALL patients undergoing allo-SCT, presence of centromeric B and absence of telomeric B genes in the donor were independently associated with a reduced risk for relapse [35]. Additionally, an increased EFS was observed by Oevermann et al. in pediatric ALL patients, who received T-cell-depleted haploidentical HSCT from KIR B haplotype donors [56]. The relevance of NK cells in the graft and in early immune reconstitution following HSCT was shown by a study identifying the alloreactive NK cell subset as having the highest antileukemia activity [57]. In our cohort, maternal NK cells did not show superior killing of their child’s leukemic blasts compared to paternal NK cells (Figure 6A). Further, persistence or absence of FM did not affect maternal NK cell killing of the leukemic blasts (Figure 6B). Thus, the observed improved outcome in hHSCT from FM^+^ mothers [4,5] cannot be attributed to direct NK cell cytotoxicity, but rather results from an interplay of different immune effector cells. Possibly, NK cells play another role by secretion of soluble factors, as NK cells from FM^+^ mothers showed a significantly higher degranulation after co-culture with their child’s leukemic blasts compared to NK cells from FM^−^ mothers (Figure 5B). 

Taken together, we identified KIR patterns that correlated with tolerance to filial cells and higher levels of FM. Although NK cells from FM^+^ mothers were more reactive against leukemic blasts of their children than NK cells from FM^−^ mothers, they failed to overcome resistance mechanisms of these malignant cells in vitro. Future studies are needed to address the influence of FM on other cell populations in haploidentical stem cell transplantation.

## 5. Conclusions

We identified three maternal genetic parameters favoring FM: (i) absence of KIR2DL3, (ii) presence of an HLA-C1 allele and (iii) a mother–child HLA-C match. Persisting FM was associated with stronger activation of maternal NK cells by the leukemic blasts of their children, but this did not translate into improved lysis of these malignant cells.

## Figures and Tables

**Figure 1 biomedicines-10-00603-f001:**
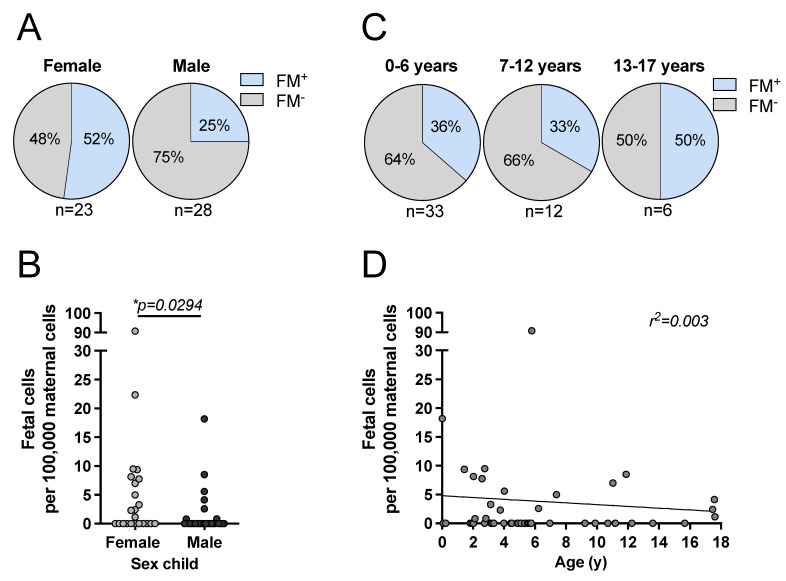
Correlation of child’s sex and age on the occurrence with the level of FM cells. FM was determined in mothers using ddPCR. The frequency of microchimeric cells was compared between mothers grouped depending on their children’s (**A**) sex, (**C**) age, and the level of FM was compared within the groups (**B**,**D**). Statistical analyses were performed using (**A**) Fisher’s exact test, (**B**) Mann–Whitney test, (**C**) Chi^2^ test and (**D**) a simple linear regression analysis. Significance level * *p* < 0.05.

**Figure 2 biomedicines-10-00603-f002:**
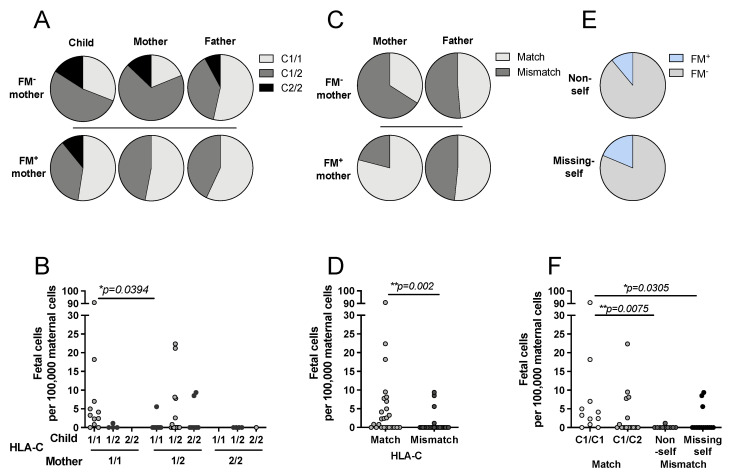
FM is favored in HLA-C matched mother–child pairs. Frequency and level of FM were analyzed in regard to filial or parental HLA-C genotypes. Child, mother, and father were grouped according to their HLA-C genotype into C1 or C2 homozygous or C1/C2 heterozygous. (**A**) Frequencies in child, mother and father grouped into FM^+^ and FM^−^. (**B**) Level of FM in mother–child pairs grouped depending on their HLA-C genotype. (**C**) Frequencies of HLA-C match or mismatch between parent and child were compared in the FM^+^ and FM^−^ groups. (**D**) Level of FM was compared between matched or mismatched mothers. (**E**) Frequencies of FM^+^ and FM^−^ mothers, grouped depending on their mismatch either in non-self or missing-self. (**F**) Level of FM in all mothers grouped into match or mismatch; the mismatch cohort was further subdivided into non-self or missing-self. Criteria for grouping into HLA-C match and mismatch are depicted in Appendix A. FM was determined in mothers using ddPCR. Exact n values are provided in Appendix A. Statistical analysis of frequencies was performed using Fisher’s exact test for two groups or Chi^2^ for more than two. Mann–Whitney test or Kruskal–Wallis test followed by Dunn´s multiple comparison test was used to analyze differences in FM levels. Statistical significance * *p* < 0.05; ** *p* < 0.01.

**Figure 3 biomedicines-10-00603-f003:**
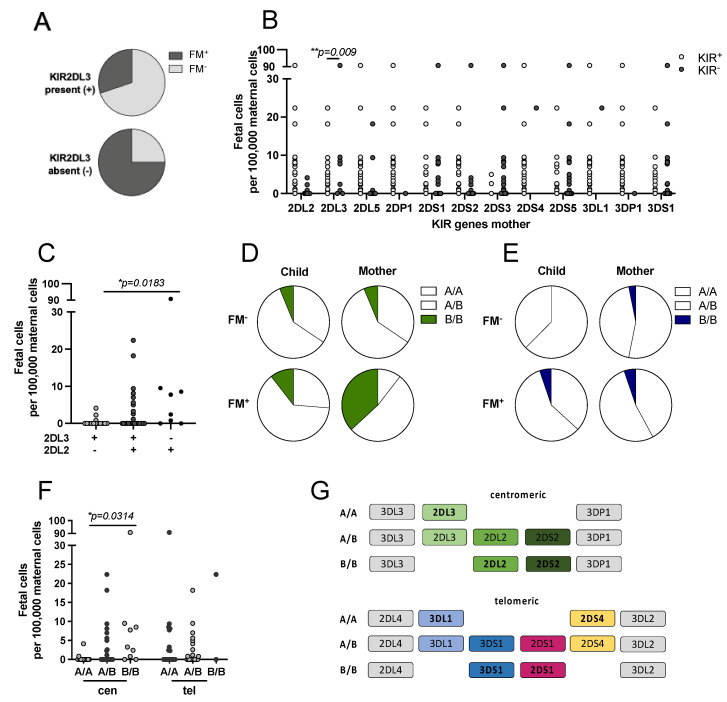
Influence of KIR genes on the occurrence and level of FM. (**A**) Frequency of KIR2DL3 gene presence or absence in FM^−^ and FM^+^ mothers. (**B**) Mothers were KIR genotyped and grouped depending on presence or absence of KIR genes. Level of FM was compared in KIR positive and KIR negative groups. (**C**) Mothers were grouped depending on their KIR2DL2 and KIR2DL3 combinations and the level of FM was compared between the groups. (**D**,**E**) Child and mother were grouped depending on their centromeric and telomeric KIR genes and the presence or absence of FM. (**D**) Centromeric motif and (**E**) telomeric motif frequencies classified as depicted in (**G**). (**F**) Influence of maternal centromeric and telomeric KIR gene regions on the amount of microchimeric cells. (**G**) Illustration of the centromeric and telomeric KIR gene regions. Exact *n* values are provided in Appendix A. Statistical analysis of frequencies was performed using Fisher’s exact test for two groups or Chi^2^ for more than two. Mann–Whitney test was used for comparison of two groups; Kruskal–Wallis test followed by Dunn´s multiple comparison test was used for more than two groups. Statistical significance * *p* < 0.05; ** *p* < 0.01.

**Figure 4 biomedicines-10-00603-f004:**
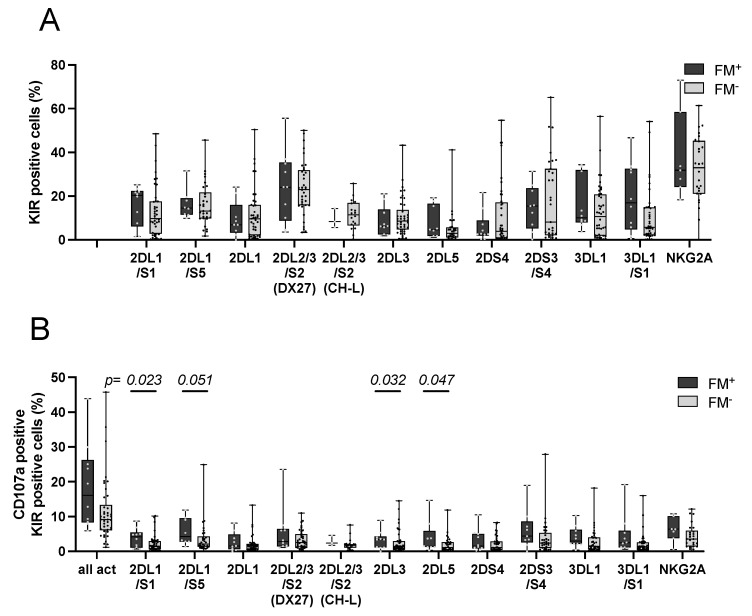
KIR expression in FM^+^ and FM^−^ mothers after co-culture with leukemic blasts. Maternal NK cells were isolated and IL-2/IL-15 pre-activated. The next day, cells were co-cultured with filial blasts in an E:T of 1:1 for four hours with addition of CD107a. After incubation, cells were stained with anti-KIR antibodies to determine the NK cell KIR phenotype. Mothers were grouped into FM^+^ and FM^−^. (**A**) All cells positive for the respective KIR+, (**B**) CD107a+ activated cells of the respective KIR. For KIR2DL2/L3/S2, two different clones were used: DX27 and CH-L. FM^+^ *n* = 3–10, FM^−^ *n* = 31–42. Depicted are box and whiskers from min to max with each point representing one sample. Statistical analysis was performed using Mann–Whitney U test for comparison of FM^+^ to FM^−^ groups. Significance level *p* < 0.05.

**Figure 5 biomedicines-10-00603-f005:**
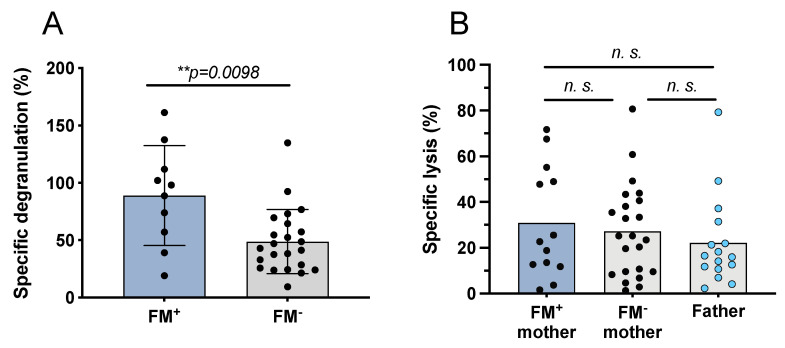
Specific degranulation and lysis of activated NK cells of FM^+^ and FM^−^ mothers after co-culture with leukemic blasts or K562. (**A**) Maternal NK cells were isolated and IL-2/IL-15 pre-activated overnight. The next day, cells were co-cultured with filial leukemic blasts or K562 in an effector to target cell ratio (E:T) of 1:10, for four hours. Mothers were grouped into FM^+^ and FM^−^. Specific degranulation of NK cells against filial blasts was calculated considering degranulation against K562 as maximum degranulation. (**B**) Analysis of maternal and paternal NK cell killing of leukemic blasts (E:T 10:1). Influence of a persisting FM (FM^+^) in mothers on NK cell cytotoxicity against filial blasts compared to fathers. Statistical analysis was performed using Wilcoxon test for paired samples and Mann–Whitney test for unpaired samples. Significance level ** *p* < 0.01, n. s. not significant.

**Figure 6 biomedicines-10-00603-f006:**
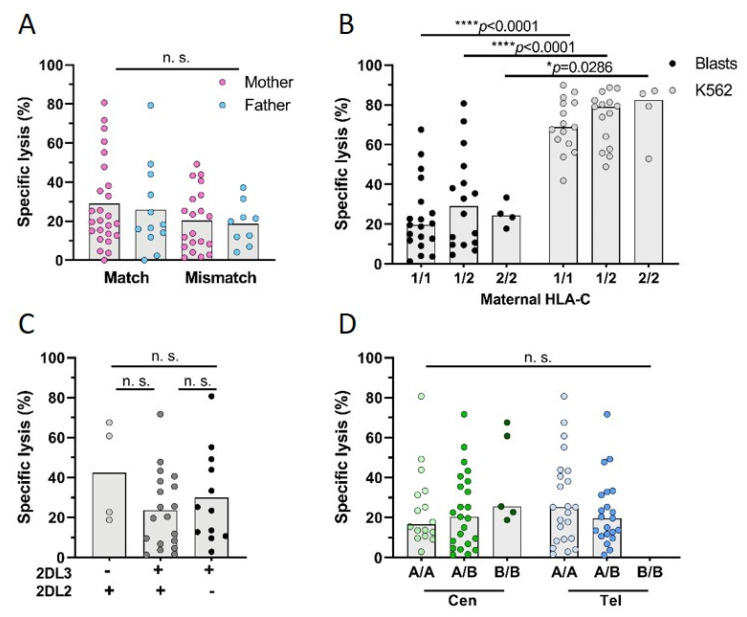
Parental cytotoxicity against filial leukemic blasts: detailed analysis of subgroups. Parental NK cells were isolated and IL-2/IL-15 pre-activated overnight. The next day, their cytotoxicity was analyzed against filial blasts and the control cell line K562 in a FACS-based killing assay. (**A**) Analysis of maternal and paternal NK cell killing of leukemic blasts (E:T 10:1) grouped into HLA-C match or mismatch between child and either mother or father. (**B**) Mothers were grouped depending on their HLA-C genotype and lysis of blasts or K562 was analyzed. Influence of (**C**) maternal KIR2DL2 and KIR2DL3 genotype or (**D**) maternal centromeric (cen) or telomeric (tel) KIR gene motifs on maternal NK cells’ specific lysis of leukemic blasts. Each dot represents one sample, bars show mean of the group. Exact n values are provided in Appendix A. Statistical analysis was performed using Mann–Whitney test for comparisons of two groups and Kruskal–Wallis test followed by Dunn’s multiple comparison test for more than two groups. Statistical significance * *p* < 0.05; **** *p* < 0.0001, n. s. not significant.

**Table 1 biomedicines-10-00603-t001:** Patient characteristics. Others include: T cell non-Hodgkin lymphoma (T-NHL), neuroblastoma (NB), hemophagocytic lymphohistiocytosis (HLH), mucopolysaccharidosis type I (MPSI), and Immune dysregulation polyendocrinopathy enteropathy X-linked syndrome (IPEX). c-ALLs are a subgroup of B-ALLs and considered separately. Depicted are counts and frequencies in the subgroups. FM determination was performed using ddPCR.

		N	%
Patients, N		70	
Sex	Female	35	50
	Male	35	50
Age (years)	Median (range)	5.72 (0.02–17.60)	
Disease	c-ALL	35	50
	T-ALL	8	11
	B-ALL	13	19
	AML	2	3
	Others	12	17
FM determination possible		51	73
	FM^+^	19	37
	FM^−^	32	63

**Table 2 biomedicines-10-00603-t002:** Correlation of genetic factors in mother, child, and father with the occurrence of FM. Samples were grouped depending on age, sex, disease, centromeric or telomeric KIR genes, KIR gene content and HLA-C genotype. Differences in FM^+^ and FM^−^ groups were analyzed using Fisher’s exact test for two groups or Chi^2^ for more than two groups followed by Cramer’s V to determine the strength of association (0.1 small effect, 0.3 medium effect, 0.5 large effect). Significance level *p* < 0.05.

		N	Value	*p*-Value	Cramer’s V
Child	Age	51	45.975	1	1
	Sex	51	3.989	0.080	0.280
	Disease	51	9.223	0.088	0.457
	Centromeric KIR	51	0.716	0.143	0.104
	Telomeric KIR	51	4.127	0.105	0.289
	KIR content score	51	4.660	0.148	0.308
	HLA-C	51	2.215	0.298	0.212
Mother	Centromeric KIR	51	8.551	0.011	0.422
	Telomeric KIR	51	0.944	0.702	0.112
	KIR content score	51	6.595	0.072	0.372
	HLA-C	51	6.979	0.031	0.387
	HLA-C match/mismatch	51	9.188	0.009	0.434
Father	Centromeric KIR	17	0.032	1	0.044
	Telomeric KIR	18	0.692	1	0.177
	KIR content score	18	1.313	1	0.226
	HLA-C	20	0.672	1	0.169
	HLA-C match/mismatch	20	0.020	1	0.032

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
