# Peer review of "Influence of Fetomaternal Microchimerism on Maternal NK Cell Reactivity against the Child’s Leukemic Blasts"

_biomedicines, 2022, doi:10.3390/biomedicines10030603_

Round 1
Reviewer 1 Report
The study “Influence of fetomaternal microchimerism on maternal NK cell reactivity against leukemic blasts in haploidentical stem cell transplantation” by Lena-Marie Martin, Anne Kruchen, Boris Fehse, Ingo Müller is dedicated to the investigation of the role that NK cells have in maintaining FM and the effect of FM on the reactivity of maternal NK cells against leukemic blasts of their children. The authors analyzed the effect of maternal and filial HLA/KIR genes on the incidence and level of FM and the impact of persisting FM on maternal NK 79 cell KIR phenotype. The functionality of maternal NK cells against leukemic blasts was assessed in vitro.
To my opinion, this article is well written, comprehensible, the methodology is well described and I suggest to accept it in the present form.
Author Response
The authors thank the reviewer for the effort as well as the positive feedback.
Reviewer 2 Report
The paper by Martin et al describes the maternal-fetal chimerism in women with children with cancers and its effect according to several parameters.
The authors describe maternal-fetal chimerism as a function of the age and sex of the child, the influence of the HLA-C locus on maternal-fetal chimerism, the role of maternal-fetal chimerism on the NK and KIR phenotype and finally the role of maternal-fetal chimerism on the reactivity of NK cells.
The paper is very interesting and well written.
They report how the HLA-C match, the HLAC1 allele presence and the absence of KIR2DL3 are factors related to higher fetal-maternal chimerism.
Questions and clarification
- How can the author explain the lack of lytic activity on leukemia blasts while degranulation of NK cells is higher in FM+ mothers?
- Please discuss, check and report if mother-child relationship was truly 3/6, in other words did the mother and the father were truly different and do not share any HLA-C or B locus?
- Could the author report the demographics of the mothers, such as age, number of pregnancies, number of transfusion(s) if any.
- I suggest to add fig s1, s2 and s3 along the text and not as supplementary materials
Author Response
- How can the author explain the lack of lytic activity on leukemia blasts while degranulation of NK cells is higher in FM+ mothers?
We tried to discuss this observation in lines 360 to 363 and hypothesized that this might bedue to restistance mechanisms of the leukemic blasts against lytic granules released by attacking NK cells. However, the sample number is too small to identify which of the candidate mechanisms contribute to which amount in these human material. We agree, that this is one of the next step to focus on in the next experiments.
- Please discuss, check and report if mother-child relationship was truly 3/6, in other words did the mother and the father were truly different and do not share any HLA-C or B locus?
We have focused on HLA-C and reported the PCR-results according to the Cooley model in the supplemtary tables S2 and S3. However, the data set waslimited by the data protection policy ot the ethics board to the necessary minimum. In appreciation of the reviewer's concern, we changed the manuscript title accordingly and removed "haploidentical stem cell transplantation".
- Could the author report the demographics of the mothers, such as age, number of pregnancies, number of transfusion(s) if any.
We have no data on mother's age and the number of (prior) pregnancies. However, the set of individually tested PCR primers assured determination of DNA from the index patient and not from sibligs.
- I suggest to add fig s1, s2 and s3 along the text and not as supplementary materials
The author's would like to leave this issue up to the editorial and ask whether there is space to include the figures in the main manuscript.